# Navigating the river(s) of systems change: a multi-methods, qualitative evaluation exploring the implementation of a systems approach to physical activity in Gloucestershire, England

James Nobles [1,2] Charlotte Fox,[3] Alan Inman-Ward,[4] Tom Beasley,[4] Sabi Redwood [1,2] Russ Jago,[1,2,5] Charlie Foster[5]

For numbered affiliations see end of article.

**Correspondence to**
Dr James Nobles;
james.nobles@bristol.ac.uk

## ABSTRACT

**Objectives** Systems approaches aim to change the environments in which people live, through cross-sectoral working, by harnessing the complexity of the problem. This paper sought to identify: (1) the strategies which support the implementation of We Can Move (WCM), (2) the barriers to implementation, (3) key contextual factors that influence implementation and (4) impacts associated with WCM.

**Design** A multi-methods evaluation of WCM was completed between April 2019 and April 2021. Ripple Effects Mapping (REM) and semi-structured interviewers were used. Framework and content analysis were systematically applied to the dataset.

**Setting** WCM—a physical activity orientated systems approach being implemented in Gloucestershire, England.

**Participants** 31 stakeholder interviews and 25 stakeholders involved in 15 REM workshops.

**Results** A white-water rafting analogy was developed to present the main findings. The successful implementation of WCM required a facilitative, well-connected and knowledgeable guide (ie, the lead organisation), a crew (ie, wider stakeholders) who's vision and agenda aligned with WCM's purpose, and a flexible delivery approach that could respond to ever-changing nature of the river (ie, local and national circumstances). The context surrounding WCM further strengthened and hampered its implementation. Barriers included evaluative difficulties, a difference in stakeholder and organisational perspectives, misaligned expectations and understandings of WCM, and COVID-19 implications (COVID-19 also presented as a facilitative factor). WCM was said to strengthen cohesion and collaboration between partners, benefit other agendas and policies (eg, mental health, town planning, inequality), and improve physical activity opportunities and environments.

**Conclusions** This paper is one of the first to evaluate a systems approach to increasing physical activity. We highlight key strategies and contextual factors that influenced the implementation of WCM and demonstrate some of the wider benefits from such approaches. Further research and methodologies are required to build the

## STRENGTHS AND LIMITATIONS OF THIS STUDY

⇒ A multi-methods, innovative approach was used to comprehensively answer the evaluative aims of this research.
⇒ Ripple Effects Mapping (REM) was used to evidence wider impacts of WCM, but also to inform the interviewee sampling and questioning.
⇒ The data collection approach was predominantly completed virtually due to COVID-19 restrictions which lessened the utility of REM.

evidence base surrounding systems approaches in Public Health.

## INTRODUCTION

Global data suggest that a quarter of adults and three-quarters of adolescents do not engage in the recommended amounts of physical activity.[1 2] National data in England paint a similar picture. Physical activity behaviour is socially patterned, and people who live in more deprived areas and who are from ethnically diverse groups have lower levels of physical activity.[3] These low physical activity levels can contribute towards poor health and well-being outcomes,[4–6] as well as having notable direct and indirect health and social costs.[4 7]

The WHO recently explored the complex drivers of low physical activity.[8] They argued that the population levels of physical activity are the outcome of an interconnected, interdependent, and evolving web of causal factors—in other words, a complex adaptive system. They outlined 45 causal factors that spanned multiple themes; from individual factors (eg, motivation, self-efficacy, family support), to sociopolitical factors (eg,

BMJ

employment status, socioeconomic status, disposable income) to transport and environmental factors (eg, public transport, air quality, walking infrastructure).[8] Others, including a 2021 commission by the Lancet, corroborate these arguments.[6 9–13]

Historically, interventions to increase physical activity have largely focused on individual agency and decision making as the primary mechanism of change.[14] These interventions seek to change individual behaviours, either through short-term educational approaches or through the provision of physical activity opportunities.[15] These approaches are also politically appealing because they require minimal political will to implement, they are relatively low cost, they are easy to measure and there is an evidence base that can be used to justify their delivery.[16–21] However, individual-level interventions can exacerbate, rather than reduce, health inequalities and seldom change the environments that people move and function within.[16 18 22]

Approaches that require lower levels of individual agency have the potential for broader reach, and more importantly, reducing inequalities.[16] A systems approach aims to influence multiple parts of the system concurrently by drawing on the expertise and resources of individuals, communities, organisations and sectors.[6 23 24] Systems approaches have the potential to reshape the environments that people interact with, and in doing so, can change default and automatic behaviours.[11 14 16 23 25] These approaches can adapt over time in response to dynamic contexts, the engagement from stakeholders and the observed impacts (intended or unintended) across the system.[6] Therefore, no two systems approaches will look, function or feel the same. There is broad consensus that systems approaches are required to help address many of the complex public health issues being faced.[8 11 19 24 26–29]

Practice has now outstripped the research and evidence base. During the last 10 years, there has been a groundswell in the implementation of systems approaches.[27 30] For example, Sport England—a national non-departmental government organisation—has allocated over £100 million for local delivery pilots. The purpose of the local delivery pilots are to test out novel, systemic and joined up approaches that have the long-term ambition of improving population physical activity, and health, outcomes.[29] The Office for Health Improvement and Disparities (formerly Public Health England) has also invested substantial resource to support local government authorities to implement a whole systems approach to obesity.[26 31] Similar patterns and investments have been observed internationally.[27 30] Despite the growing interest in, and implementation of, systems approaches in Public Health, there is a distinct lack of published research surrounding their evaluation.[19 27 30 32 33] In this paper, we will provide the results from one such evaluation.

We Can Move (WCM) began its implementation in April 2018. Through a systems approach, it aimed to influence the physical activity levels across the Gloucestershire population, with a specific focus on populations with very low levels of activity (less than 30 min per week). Active Gloucestershire is a voluntary sector organisation that co-ordinated WCM, otherwise known as the 'backbone organisation'.[34] Together, with a small number of stakeholders from other organisations, they made up the WCM implementation team. The implementation team worked with a wide variety of community groups, organisations and system leaders from multiple sectors to overcome the barriers facing individuals (eg, opportunities for physical activity), communities (eg, the quality of our physical and social environments) and the wider system (eg, the policies and people that make the system work as it does). It was anticipated that WCM would not only benefit the physical activity levels in Gloucestershire, but many other associated agendas such as climate change, overweight and obesity, and mental health. Table 1 provides further details about WCM.

In April 2019, Active Gloucestershire commissioned the National Institute for Health and Care Research Applied Research Collaboration West (NIHR ARC West) to evaluate WCM. This paper will: (1) identify the strategies which supported the implementation of WCM, (2) highlight the barriers to WCM being implemented, (3) illustrate key contextual factors that influenced its implementation (ie, factors that are not part of the intervention per se or were already established pre-implementation[35 36]) and (4) demonstrate the range of impacts associated with WCM being implemented. The WCM programme continued beyond the evaluation period.[37 14]

## METHODS
### Study design
This study draws on the data from a multi-methods evaluation of the WCM programme. Further details of the whole evaluation are available elsewhere.[37] Here, we will report on the findings from a set of stakeholder interviews and Ripple Effects Mapping (REM) workshops. In brief, REM is a participatory, qualitative method for understanding and visualising the wider intended and unintended impacts of a project or programme (described in depth later and elsewhere[38]). The data pertaining to these two methods were drawn on here as they provide in depth insights into how WCM functions, the contextual factors which influence its implementation, and to highlight an array of impacts associated with WCM. These two qualitative methods are complimentary, and the findings can be triangulated to comprehensively address the research questions.

It is important to note here that the lead researcher (JN) was embedded in the Active Gloucestershire organisation for one day per week. As part of this role, they attended programme-related meetings and workshops, met with wider stakeholders, and consequently, created meaningful relationships with the implementation team. The embedded nature of this role enabled the researcher

**Table 1** Description of We Can Move (WCM)

| | |
|---|---|
| **About WCM**: in 2017, Active Gloucestershire secured funding to research alternative approaches to improving the population levels of physical activity. during this time, they developed a high-level Theory of Change for a new approach, initially named 'Gloucestershire moves'. The Theory of Change included systems science, behaviour change principles and social movement building (for further information, see Nobles *et al*).[37] It would see methods like systems mapping being used to understand the complexity of the problems being faced locally, and then models such as the Behaviour Change Wheel[14] helping to inform the development of interventions.<br><br>During 2020,[37] the Gloucestershire Moves changed its name to 'We Can Move'.<br><br>The vision for WCM was to inspire, connect and enable individuals, communities and organisations to help the least active, move more.<br><br>It hoped to achieve this by engaging a wide variety of community groups, organisations and system leaders to remove the barriers facing individuals (eg, the opportunities for physical activity), communities (eg, the quality of our physical and social environments), and the wider system (eg, the policies and people that make the system work as it does). | The Theory of Change was initially tested on the Falls Prevention Initiative and the Community-based Initiative (the two case studies being evaluated). The Theory evolved slightly throughout this piloting, before being phased into all of Active Gloucestershire's work between 2018 and 2021.<br><br>WCM cost an estimated £3 000 000 to implement between April 2018[37] and April 2021.<br><br>Further information is available in Nobles *et al*.[37]<br><br>**About Active Gloucestershire**: Active Gloucestershire are a small voluntary sector organisation funded through national and local grants, local public sector organisations, and charitable donations. Within WCM, Active Gloucestershire's role was to facilitate this approach—also known as the 'backbone organisation'. Prior to WCM, Active Gloucestershire's role was to deliver physical activity sessions and other types of interventions to the public. This often meant working with those who were already physically active.<br><br>**Setting**: WCM was implemented in Gloucestershire, England. Population of approx. 640 000 people. Demographic is ageing, 95% white British, and has areas of high deprivation. An estimated 130 000 adults do less than 30 min physical activity per week, and 52.7% of children do less than 6 hours of activity per week. |

to understand the nuance of the WCM programme, and subsequently, was seen to bolster the evaluation effort.[39]

Further details related to the methodology is available in online supplemental file 1 (COREQ checklist).

## Case studies

Three aspects of WCM were identified to create case studies which would help to explore how the Theory of Change (see table 1) was implemented in practice, and moreover, how each case study contributed towards the WCM programme. These case studies provided artificial and pragmatic boundaries around a complex, messy programme that would otherwise be unwieldy to evaluate within the resource constraints.

Two case studies focused on smaller projects situated within WCM (Falls Prevention Initiative and a Community-based Initiative) and the last explored, more broadly, how WCM engaged with organisations and wider stakeholders around physical activity (table 2). Each case study was identified and agreed on through discussions between the research team and Active Gloucestershire. An adaptive evaluation framework was applied to each case, meaning that different methods could be applied as needed to best answer the evaluation questions. At the same time, some methods, such as semi-structured stakeholder interviews and REM, were applied across all studies to ensure consistency in the type of data collected and insights generated. This further strengthened the rationale to centre this paper on the data and results related to interviews and REM.

## Data collection

We focus here on the data gathered using semi-structured interviews and REM. We planned to recruit 10 stakeholders per case study to participate in a semi-structured interview (n=30 total). This sample size was informed by the resources available and the need to reflect a balance of views across the case studies. Prospective interviewees were purposefully sampled to gather a diverse range of perspectives via three main channels: (1) through conversations with the WCM implementation team (ie, those involved in the design and delivery of WCM, predominantly Active Gloucestershire staff); (2) using the REM outputs and (3) using data gleaned from an annual stakeholder survey. The survey was developed by the evaluation team to understand the levels of engagement and involvement of wider stakeholders in WCM. Thirty-seven stakeholders completed this survey in 2020 and 2021. We purposefully sampled interview participants based on their responses to elicit maximum variation in the data set. In summary, interviewees were either involved in, or affected by, the implementation of WCM (or the specific case study within WCM).

A topic guide (online supplemental file 2) was used to provide consistency within the stakeholder interviews. It paid attention to how stakeholders were involved in WCM, key contextual factors, the strategies facilitating and the barriers inhibiting its implementation and the perceived impacts associated with WCM. Probing questions were then tailored to each interviewee based on

| **Table 2** Case descriptions | | |
|---|---|---|
| **Case 1: Falls Prevention Initiative** | **Case 2: Community-based Initiative** | **Case 3: WCM movement building** |
| **Background**: Gloucestershire has an ageing population. Older adults are at a greater risk of falling and being admitted to hospital due to falling.<br><br>**About**: A healthcare organisation funded Active Gloucestershire to develop a Falls Prevention Initiative for the county, with the stated outcome of reducing falls admissions to the Accident and Emergency department.<br><br>Active Gloucestershire worked with community members and professionals to design and develop Fall-Proof. Behaviour change techniques were used to design a behaviourally informed intervention.<br><br>Fall-Proof aimed to increase older adults' awareness of falls risk, to encourage strength and balance exercises at home, and to increase referrals to community-based strength and balance classes. | **Background**: Barton and Tredworth is a small, culturally diverse, deprived area in Gloucester.<br><br>About: Active Gloucestershire wanted to start working in a hyperlocal manner that responds to, and works with, residents.<br><br>As part of WCM, Active Gloucestershire received funding from a national organisation to work with the local community, including a group of local women, in Barton and Tredworth. These women wanted to create opportunities for other women, particularly Muslim women, to participate in physical activity.<br><br>This group organised many opportunities for physical activity, linked in with local organisations, and championed physical activity in their communities. Active Gloucestershire facilitated the group. | **Background**: Social movement building is a core component of the WCM Theory of Change. It provides a framework for stakeholders and organisations to integrate into, and take ownership of, WCM.<br><br>**About**: WCM seeks to work with multiple organisations and sectors to make Gloucestershire a more conducive place for physical activity. WCM encourages stakeholders to see their place in the system, to see how they could contribute to making physical activity the norm, and to see that many different agendas would benefit from improved collective working.<br><br>Active Gloucestershire supported and facilitated WCM's implementation, alongside providing opportunities for organisations to engage in and support WCM, and engage with each other. |
| WCM, We Can Move. | | |

content discussed and the REM output which they may be associated with (further information in table 3). For example, an interviewee may have been asked how their work contributed towards an impact that had been recorded in an REM output. Interviews were conducted by the same researcher (JN) using online videoconferencing platforms. Interviews typically lasted 30–60 min.

Regarding REM, we adapted the original in-depth REM technique from Chazdon et al[40] so that it captures the temporal aspects of systems change efforts. The adaptations to this method have been reported elsewhere.[38] In brief, data were gathered via participatory workshops with the implementation team, and were led by an experienced facilitator (JN). The facilitator guided participants through a series of activities (table 3) that created a visual REM output which documented the actions and activities undertaken, along a timeline, and their associated initial and wider impacts.

The first REM workshops were conducted face to face with the whole WCM implementation team, lasting 120–150 min. The implementation team split into smaller groups to focus on particular aspects of, or a project within, WCM. For example, one group focused on the Falls Prevention Initiative and another on the Community-based Initiative. There was time within the workshops for participants to create multiple REM outputs. Due to COVID-19 restrictions, all follow-up workshops were carried out using online videoconferencing platforms with smaller groups (based on the groups they worked during the first in-person REM workshop). Follow-up workshops

(60–90 min) aimed to update previous REM outputs with new actions, activities, and impacts that occurred throughout the last 3–4 months. Data were recorded in a paper-based format in the face-to-face sessions, and data from subsequent follow-up workshops were entered into Vensim PLE, and with permission, conversations were recorded directly through online videoconferencing platforms. Further details on how we applied REM in the WCM evaluation is available in a methodological paper.[38] See figure 1 for a simplified and illustrative REM output.

Data from the two methods were collected between May 2019 and March 2021. The REM workshops were generally conducted prior to the participant interviews so that the outputs could inform the interview questions. We completed the interviews first for the Community-based Initiative by May 2020, (first and last REM session: May 2019 and July 2020), then the Falls Prevention Initiative by October 2020 (first and last REM session: December 2019 and March 2021), and finally, for the WCM movement building case study by January 2021 (first and last REM session: December 2019 and March 2021).

## Data analysis

Interview recordings were transcribed verbatim and uploaded into NVivo V.12 for data management. A framework analysis approach was applied to all qualitative data within the WCM data set,[41] which included the data from interviews and the REM outputs. The initial framework was devised around core features of interest, including contextual factors, implementation strategies,

**Table 3** Summary of REM stages[38]

| REM stages | Description |
|---|---|
| 1. Introduction to REM | The facilitator gave a brief presentation to introduce the REM method. This included background to the method, the limitations of traditional evaluation methods and an outline of the approach used in the REM workshops. Not required in follow-up workshops.<br>Time recommended: 10–15 min |
| 2. Peer/team-based discussions | Participants split into pairs or small project groups and had conversations about the work they are involved in, what they are proud of, what has gone well within their work, and what has been challenging. These discussions were guided by principles from Appreciative Inquiry.[38 63] Conversations served as a warm-up to the main activity.<br>Time recommended: 10–20 min |
| 3. Mapping the impacts | Each project group was provided with a large piece of paper that had a timeline placed longitudinally across it. The timeline spanned from the beginning of WCM (April 2018) to the present time (December 2019). Participants were asked to think about the main activities or actions that were carried out throughout this timeframe, and to write these onto the map at the time which they occurred. They were then asked to map out the impacts associated with the respective activities. Arrows were draw between activities and impacts to denote the relationship. Participants were encouraged then to map any additional 'ripples' that manifest following this activity and\or impact. The facilitator assisted groups where needed. During this time, groups were able to work on REM outputs associated with two to three projects/areas of work.<br>Time recommended: 60–90 min. |
| 4. Reflecting on the impacts | Part way through the previous stage, the facilitator asked participants to consider additional questions when mapping. These included: (1) identifying intended and unintended impacts, (2) stating which organisations/stakeholders/groups were involved or affected, (3) financial implications, (4) what else may have contributed to impacts occurring, and (5) how these activities/impacts link with wider WCM work. Recurring trends may also have been apparent in their outputs. When organisations/stakeholders/groups were identified, these provided the evaluation team with potential interview participants.<br>Time recommended: Concurrent to stage 3. |
| 5. Most and least significant changes | Once most of the actions, activities and impacts are mapped out, the facilitator then asked the groups to identify what they considered to be the most-significant and least-significant changes in their output. Groups had time to discuss what may have contributed to these changes occurring as they did, and whether in retrospect, they would do anything differently. Data were not recorded for evaluative purposes at this stage.<br>Time recommended: 10 min |
| 6. Group feedback and learning | The final part of the workshop provided time for participants to feedback to the group the main aspects of their work that was mapped, what their reflections on the map were, and what they had learnt from the REM process. The facilitator guided this discussion.<br>Time recommended: Remaining time. |

REM, ripple effects mapping; WCM, We Can Move.

and intended and unintended outcomes. Data pertaining to the three case studies were analysed independently, so that similarities and differences could be drawn out. Additional content analyses[42] were used to quantify certain aspects of the REM outputs (eg, number of organisations involved, number of policies influenced etc…). More information regarding the analysis of REM outputs is available in a methodological paper.[38] All data were analysed by the lead researcher (JN) and supported by a second researcher (CFox) to minimise bias. To increase the dependability of the findings, a researcher (JN) had regular discussions with the implementation team and the wider stakeholders to ensure preliminary findings resonated with their experiences. Findings, not data, were triangulated from the interviews and REM as part of the framework analysis for the purpose of this paper. The majority of the findings related to (1) reasons for involvement, (2) implementation strategies, (3) barriers to implementation and (4) contextual factors originate from the analysis of the interviews, bolstered by the REM analysis. The findings regarding the impacts and outcomes predominantly arise from the REM analysis and strengthened by the interview findings.

### Patient and public involvement
No patient and public involvement.

### RESULTS
Thirty-one interviews were conducted across the three case studies. Interviewees represented local government authorities, healthcare services and commissioning, voluntary and community sector organisations, local elected members, community residents and Active

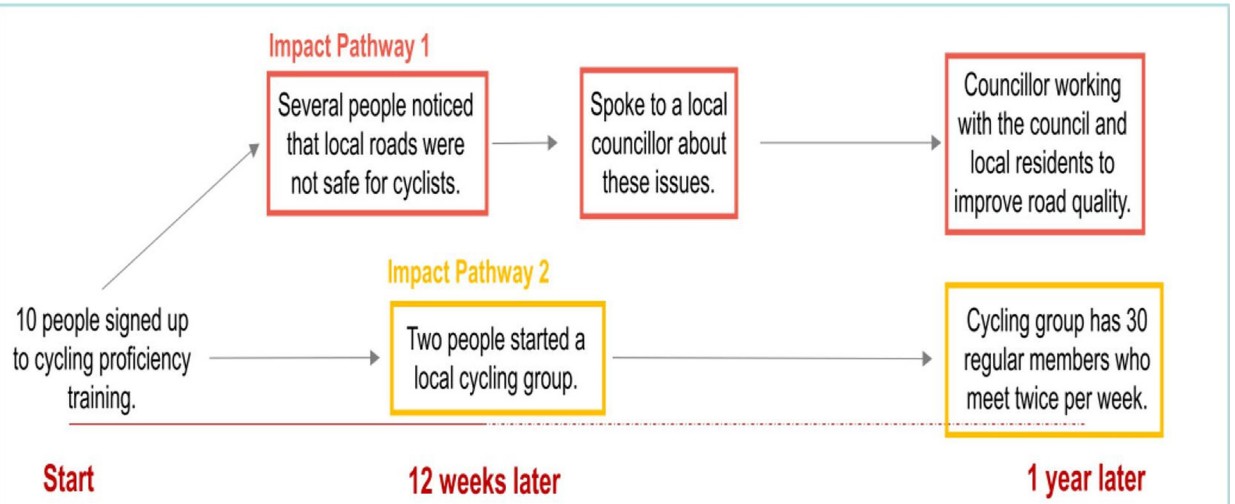

**Figure 1** Simple REM example. REM, Ripple Effects Mapping.

Gloucestershire staff. Fifteen REM outputs were created with the input of 25 participants (n=17 implementation staff, n=8 community residents involved in the Community-based Initiative). Seven outputs were updated over the course of a year (further information available in Nobles *et al*[38]). One output related to the Falls Prevention Initiative and Community-based Initiative, respectively, and the remaining outputs were included within the third case study (WCM movement building). The final themes from the analysis of the interview and REM outputs are presented under five key headings: (1) reasons for stakeholder involvement; (2) implementation strategies; (3) barriers to implementation; (4) contextual factors and (5) impacts and outcomes (table 4). Text in bold below represents a subtheme.

We have developed a white-water rafting analogy to help explain, and interconnect, many of the findings below. The analogy illustrates how an approach like WCM (represented by the raft) requires a guide (ie, Active Gloucestershire), a crew (ie, wider stakeholders) and the ability to negotiate the dynamic nature of the river (ie, the broader context and circumstances which influence implementation).

### Reasons for involvement

Stakeholders frequently had multiple reasons for getting involved in WCM. Across the three cases, stakeholders believed in the **co-benefits** associated with WCM and physical activity (eg, mental health, social connections, health inequalities), which could be realised by individuals, communities, organisations and wider sectors (eg, Transport or Social Care). Many also noted that WCM aligned with the **strategic priorities of their own organisation**, and so their involvement in WCM would be mutually beneficial. For example, one stakeholder highlighted how WCM could benefit a council agenda around adverse childhood experiences (ACEs):

ACEs is one of the sort of Health and Wellbeing Board priorities and it has been a priority since 2018. Part

of that and the research around ACEs was around, we know that you can build resilience through regular sports participation and we're using the five ways to wellbeing as our sort of resilience building tool. – Participant 2, Case 3, County council public health

Other reasons were more specific to individual cases. Some bought into the Falls Prevention Initiative because they believed it would bring about 'quick-wins'—most notably, reductions in the number of people being admitted to hospital because of falling. On the other hand, in the Community-based Initiative, several community residents became involved because they lived locally and wanted to benefit the community, while others wanted to learn about the WCM Theory of Change (table 1).

### Implementation strategies

Our white-water rafting analogy is particularly useful here to present the three main themes. These include: (1) the role of the guide (ie, the backbone organisation), (2) developing the right crew (ie, wider stakeholders) and (3) the ability of WCM to 'read the river' (ie, local context and circumstances). These factors are interconnected and interdependent, as explained below, using the white-water rafting analogy.

A fundamental **role of a guide** is to steer the raft by talking to, motivating, and inspiring the crew. In WCM, Active Gloucestershire held the role of the guide, also referred to as the 'backbone organisation'. While the backbone role varied across the three case studies, its importance was widely acknowledged by participants. For example, in the Community-based Initiative (case 2), the backbone role was held by one staff member of Active Gloucestershire. Interviewees stated that they were pivotal in that they facilitated relationships between stakeholders, that they provided knowledge and expertise (about physical activity and about local networks of stakeholders and organisations), and that their role—and the way in which they worked—was 'more than a job' (participant 6, case 1, community member). In the context of WCM (case 3),

**Table 4** Framework analysis

| | Case 1: Falls Prevention Initiative* | Case 2: Community-based Initiative† | Case 3: WCM movement building‡ |
|---|---|---|---|
| 1. Reasons for involvement | ▲ Project aligned with, and could help, organisational priorities. <br> ▲ Belief in the project and the associated benefits. <br> ▲ Opportunity to network and share learning across organisations. <br> ▲ Perceived as a 'quick win'. | ▲ Experienced, or belief in, benefits of physical activity. <br> ▲ Want to increase physical activity opportunities and levels. <br> ▲ Believe in the wider benefits to society and the community (ie, 'not just about physical activity'). <br> ▲ Opportunity to learn about new approaches. <br> ▲ Live in the local area. | ▲ Shared priorities, passions and values. <br> ▲ Personal, organisational or sectoral benefits from engagement. <br> ▲ Contractual obligations. |
| 2. Implementation strategy | ▲ Backbone role: <br> – Links with community. <br> – Signposting to materials. <br> – Marketing expertise. <br> – Bringing people together. <br> ▲ Key stakeholders involved in design of the initiative. <br> ▲ Capitalise on pre-existing relationships. <br> ▲ Dissemination via influential stakeholders and residents. <br> ▲ Formal conduits between organisations. <br> ▲ Behaviour Change Wheel approach to intervention design. <br> ▲ COVID-19 causing statutory organisations to consider alternative ways of working. | ▲ Backbone role: <br> – Pivotal mechanism. <br> – Facilitative. <br> – Provide knowledge, expertise and credibility. <br> – 'A way of working'. <br> ▲ Steering group role: <br> – Understand local needs and wants. <br> – Tailored, non-judgemental meetings. <br> – Proactive and relatable community members. <br> – Central hub to nurture ideas. <br> – Disseminate information to community about physical activity and related opportunities. <br> – Champion physical activity for community. <br> ▲ Engage the right people and organisations. <br> ▲ Create a collective and shared vision. <br> ▲ Establish effective communication channels between stakeholders. <br> ▲ Understand root causes of low physical activity, driven by data and insight. <br> ▲ 'Go with the flow'. <br> ▲ Keep it local and draw on community assets. <br> ▲ Selling physical activity through persuasive and creative communications. | ▲ Backbone role: <br> – Facilitating (and disrupting) the system. <br> – 'Joining the dots' – fostering new relationships. <br> – Linking in with, and establishing, professional networks. <br> – Influencing the system and system leaders. <br> – Imparting knowledge and expertise. <br> – Designing and delivering projects. <br> – Supporting other organisations. <br> – Fiducial responsibilities. <br> – Champion WCM and physical activity. <br> ▲ Adaptive and strong governance for WCM. <br> ▲ Developing strong foundations for WCM: <br> – Passion of Active Gloucestershire team. <br> – Informal meetings. <br> – Council and senior leadership support. <br> – Alignment with strategic priorities in the county. <br> – Funding in place. <br> ▲ Working with the right people who have a shared vision, agenda and purpose. <br> ▲ Focus on building strong relationships between people and organisations. <br> ▲ Events and networks can initiate and catalyse relationship building. <br> ▲ Develop conduits between organisations and WCM. <br> ▲ Ability of WCM to 'read the river' – timing and conditions. <br> ▲ Celebrating success. <br> ▲ COVID-19 led to new opportunities, funding and partnerships. |

Continued

**Table 4** Continued

| | Case 1: Falls Prevention Initiative* | Case 2: Community-based Initiative† | Case 3: WCM movement building‡ |
|---|---|---|---|
| 3. Barriers to implementation | ▲ Clarity on roles, responsibilities and involvement of wider stakeholders.<br>▲ Communication between backbone organisation and wider stakeholders.<br>▲ 'Demonstrating impact'—evaluation, measurement, and monitoring of complex projects.<br>▲ Managing conflicting stakeholder tensions, mindsets and ways of working.<br>▲ COVID-19 impact on delivery and participation.<br>▲ Linking the Falls Prevention Initiative to WCM.<br>▲ Backbone organisation capacity.<br>▲ Length of time to design intervention. | ▲ Overcoming bureaucracy, red tape and administrative challenges.<br>▲ Monitoring 'success' of Community-based Initiatives.<br>▲ At odds with transactional mindsets of organisations and senior leadership.<br>▲ Limited stakeholder engagement.<br>▲ Acknowledge challenges within a steering group (format, aims, opposing views). | ▲ Stakeholder understanding, communications and perceptions of WCM.<br>▲ Clashing of mindsets.<br>▲ (Continued) Engagement in a social movement.<br>▲ Inconsistent preferences surrounding evaluation.<br>▲ COVID-19 prevented progress and collaboration across WCM.<br>▲ COVID-19 impact on stakeholder engagement, and prioritisation of WCM by organisations. |
| 4. Contextual factors | Facilitative contextual factors<br>▲ History of local falls prevention work.<br>▲ Organisational strengths of Active Gloucestershire.<br>Inhibitory contextual factors<br>▲ Institutional transactional ways of working.<br>▲ Active Gloucestershire going through transition period. | Facilitative contextual factors<br>▲ Active Gloucestershire as a trusted, well-connected and established organisation.<br>▲ Established Voluntary and Community Sector and history of community building.<br>Inhibitory contextual factors<br>▲ Mindset shift occurring in some organisations.<br>▲ Negative perception towards previous approaches - '(locality was) a picked-on area'.<br>▲ Varied demographic of local community.<br>▲ Cultural norms against physical activity. | Facilitative contextual factors<br>▲ Active Gloucestershire well-connected, respected and knowledgeable organisation.<br>▲ 'WCM pushing on an open door'<br>Inhibitory contextual factors<br>▲ Demographic profile of Gloucestershire.<br>Both facilitative and inhibitory<br>▲ COVID-19 implications. |
| 5. Impacts and outcomes | ▲ Awareness, use and uptake of the Falls Prevention Initiative work.<br>▲ Quality of the Falls Prevention Initiative materials.<br>▲ Stakeholders feeling part of something bigger.<br>▲ Wider engagement in Falls Prevention Initiative materials (locally and nationally).<br>▲ Limited impact on referrals to strength and balance providers. | ▲ Perceived increases in community physical activity.<br>▲ Improved/changed physical activity infrastructure and provision.<br>▲ Enhanced community engagement, cohesion, and empowerment.<br>▲ Growth, reach and self-organisation of work.<br>▲ Influence on, and buy-in from, wider sectors.<br>▲ New money and funds generated.<br>▲ Steering group:<br> – Group ownership changed over time.<br> – Buy-in from local councillors.<br> – Feel part of something bigger.<br> – Feel empowered and accomplished. | ▲ Successful buy-in and support for WCM.<br>▲ Mindset shifts within organisations and senior leaders.<br>▲ 'Linking up the system'—new relationships created.<br>▲ 'Systems change happening in pockets'.<br>▲ Beneficial impacts and influence on communities, organisations and sectors.<br>▲ Uptake of physical activity and interventions.<br>▲ New money and funds generated.<br>▲ New projects established.<br>▲ Other areas adopting/interested in WCM.<br>▲ Supporting, benefiting and aligning with other strategic agendas. |

*Falls Prevention Initiative data collection: 12 stakeholder interviews, one REM output (created over four iterations).
†Community-based Initiative data collection: 10 stakeholder interviews, one REM output (created over five iterations).
‡WCM movement building data collection: Nine stakeholder interviews, 13 REM outputs (five created over multiple iterations).
REM, ripple effects mapping; WCM, We Can Move.

the backbone role was held by Active Gloucestershire as an organisation. Active Gloucestershire were seen to be centrally involved in facilitating how the physical activity system was organised—from the stakeholder networks to the way in which funding was distributed within the sector. This was referred to as a 'behind the scenes' role (participant 7, case 3, local third sector organisation). Thus, they were believed to have substantial influence given their strong connections to senior leaders across Gloucestershire, further reinforced by their credibility as an organisation. Interviewees also spoke of the intricate knowledge (of people, places and communities) and extensive expertise (in physical activity, project management and behaviour change methodology) that Active Gloucestershire held and were able to share with others. A large part of the backbone role was to 'join the dots' (multiple interviewees references)—to bring people, organisations and their agendas, together. They did this via formal (eg, setting up stakeholder networks) and informal (eg, email introductions) methods. Several interviewees reported that Active Gloucestershire provided in-depth advisory support to their organisation given that they were relatively new to the county and/or work in the Public Health sphere. At their core, Active Gloucestershire were passionately championing, facilitating and co-ordinating WCM.

> I think that what Active Glos did getting all those people together, was the contacts that they've got and the networks that they've got. They're very different from our health network, so I think that was brilliant. It did make sure that the community voice from a lot of different perspectives remained through those [Falls Prevention Initiative] planning stages definitely. – Participant 3, Case 1, NHS commissioning

> I think it's imperative to have somebody on the ground that knows the local area, can build a relationship with local people, they've got that local knowledge, they can connect like-minded people and just offer that bit of support – Participant 3, Case 2, City council

To negotiate the turbulence of the river, a raft needs more than a guide, it needs a crew. This is where the implementation and success of WCM was largely hinged on its engagement with wider stakeholders. Interviewees said that Active Gloucestershire placed a lot of effort on **developing the 'right' crew**; the people and organisations to work with as part of WCM. The 'right' people and organisations typically comprised those whose vision (ie, how they work and what they are aspiring to) aligned with that of WCM, and whose agenda (ie, the things that are important /prioritised within their organisation) would benefit from the work of WCM. These two elements, vision and agenda, enabled wider stakeholders to see value in WCM and to better see their role in WCM. Furthermore, interviewees spoke about the personality and behavioural characteristics (or 'mindset'—that is, way of thinking about problems and actions) of wider stakeholders who

were influential within WCM. This included an ability to build strong relationships, to look and work beyond their own organisational priorities (and thus acknowledge that they work within a system rather than silos), and to feel comfortable working—or attempting to work—in a different, and perhaps more collaborative and equal, manner. These crew members often acted as conduits for information about WCM to flow through to their senior management, therefore, working as a strategy to increase organisational buy-in to WCM.

> I have always been that new way, I've always done this sort of way, I always think connecting the dots is much better, more effective so I've never worked in that sort of old way and that silo approach I guess. So I think there might be a bit of challenge around how they work and you know bringing people—great, get everyone in a room, get everyone interested but then you might get some people that would just automatically flip back to, right what's on priorities, what's on the agenda I've got to do today. – Participant 2, Case 3, County council public health

To identify the 'right' people, Active Gloucestershire staff invested substantial time in attending and organising networks (via REM analysis, at least 23 networks were attended or organised). These networks often led to serendipitous outcomes; for example, Active Gloucestershire staff met new people and organisations, organised meetings outside of the networks, which subsequently led to unplanned joint working that would benefit the overall implementation of WCM (and the work of the other organisation). We noted, through the REM analysis, that 82 organisations had been engaged in WCM throughout the data collection (April 2019–January 2021), and there were many examples of these unanticipated partnerships in the dataset. Most of the work, and relationships developed, in the Community-based Initiative (case 2) came about because of an initial network meeting held by Active Gloucestershire called 'Joining the Dots', which was attended by approximately 65 local organisations. However, the REM and interview analyses also suggest that time was needed to develop meaningful, trusting relationships, while the length of time required varied dependent on the stakeholder or organisations engaged with.

With the guide and crew established, the final key strategy was the ability for WCM to '**read the river**'. Interviewees described how WCM (and associated projects) had been able to adapt to local and national contexts and circumstances (see COVID-19 subtheme below), and how key stakeholders (ie, guide and crew) could react quickly to these circumstances to benefit the overall approach. For example, this meant being able to (1) develop proposals with organisations quickly to secure new funds, (2) present evaluative information to funding partners to reassure them that progress was made and (3) change governance structures to better support the vision of WCM. This need for adaptation was recognised by many of the interviewees. In contrast, the

Falls Prevention Initiative (case 1) perhaps resembled a more traditional approach to service delivery; it had key performance indicators, it was held accountable by the funding organisation to these indicators and deliverables, and the relationship between funders and the providers was considered somewhat transactional. Here, we saw a clash between traditional and the desired ways of working within WCM. WCM appeared, according to interviewees, to work more effectively when it was able to alter its course in response to the ever-changing, choppy nature of the river (ie, circumstances and context).

> It doesn't feel within the [organisation] to... there's not that trust, I don't believe, from the [organisation] for us to adapt the programme because we are seeing a need to change and adapt and I think there's not that trust of, okay, we understand that like there is in some other areas of We Can Move. I think with this [the Falls Prevention Initiative] you have to justify and evidence every change you make... – Participant 8, Case 1, Implementation team

> It's all about the relationships isn't it and you can't— the organisations you might think you are best placed to work with might not want to work with you or you might not have the relationship established. So yes you kind of have to build that don't you, in the short term it might mean doing different things or working with different partners, because as you say that's where the appetite is and the energy is. – Participant 8, Case 2, National physical activity organisation

The **COVID-19 pandemic** both foregrounded and bolstered the ability of WCM to adapt to changing contexts. While creating many challenges for the implementation of WCM, COVID-19 was also a catalyst in enabling or necessitating multisectoral, multiorganisational working. This theme was particularly present within the REM analyses. For example, in the Falls Prevention Initiative (case 1), the educational materials were disseminated more broadly than initially anticipated—and through new organisations—due to the requirement for many older adults to isolate in the early waves of the pandemic. More broadly across WCM (case 3), the impact of COVID-19 meant that many organisations had to de-prioritise their involvement in WCM to remain operational. This subsequently meant that WCM had to alter its priorities which it was able to do in part because of its adaptive governance structures (ie, the ways in which other organisations become involved). As new money became available across the system from central government and other national funding agencies, Active Gloucestershire, due to their credibility as an organisation and centrality within the local networks, were able to bring stakeholders together from multiple organisations to capitalise on these funding streams to support the health—and wider needs—of local populations. In turn, this then led to new areas of work, new relationships, and wider impacts of WCM across the system.

> I guess COVID and needing to work in a new way just made people look up a bit [i.e. look outside of their organisational silo] because there was a need. Previously you had a service and then suddenly you didn't, so there was a gap which you couldn't fill with anything traditional. Why would you look up when you've got a working service, essentially? But when that need came, people were like, 'Okay, so what can we do differently?' and then that's when those links— people were like, 'Okay, I'll ask if he's got any ideas' or... - Participant 6, Case 1, NHS commissioning

### Barriers to implementation

Four main barriers to the implementation of WCM were identified. These include challenges related to evaluation, issues surrounding opposing mindsets within WCM, misaligned understanding and expectations of WCM, and the negative implications of COVID-19.

The **evaluation** of WCM, and associated projects (eg, the Falls Prevention Initiative and Community-based Initiative), was problematic for several reasons. Across the three case studies, there were inconsistent expectations around the purpose and role of the evaluation across WCM. For example, in case 2, several interviewees stated that it was difficult to measure and capture the unanticipated impacts of a community-driven initiative (that did not have a predetermined project plan). In case 1, interviewees disagreed on whether it was important to focus on the 'hard numbers' associated with the Falls Prevention Initiative to determine the success of the project. For some, these numbers (ie, reduction in number of people being admitted to hospital due to a fall) were the most important 'success' measure, but for others, especially those closest to the implementation, believed that it was not feasible to monitor this or attribute change to the intervention directly. These issues around attribution of effect were also noted among the wider WCM stakeholders (ie, case 3).

> They [senior leaders] were literally grilling the community builders and I don't think I was meant to stand up, I was just in the gallery viewing the whole thing and I just said, 'Look, can I speak a minute?' I said, 'At the end of the day this isn't something that you can just see happening overnight, you have to wait.' It's taken however long for me to get to where I am now, 'cause I'm doing, I suppose you could say, unofficial community building, but as a volunteer, I suppose, but it takes time. If I expect everything to happen overnight, I'll just give up and won't bother and so I think back to me, this is something that you can't just assess overnight and you've got your answers, you have to wait to see the positives that come out of it. – Participant 4, Case 2, Community member

This is closely entwined with the next barrier, a **clash of mindsets** within WCM. In part, it appeared that the expectations surrounding the evaluation were linked to the

mindset and ethos of the individual, organisation and/or sector. WCM was advocating for an alternative way of working across the system, meaning that multiple organisations are involved in the implementation and adaptation of WCM over time. However, each organisation coming aboard WCM had their own ways of working, and in particular, some organisations adopted transactional ways of working. This meant that these organisations were typically accustomed to commissioning services, setting key performance indicators (thus linked to evaluation issues above), having regular performance update meetings, and therefore, having a hierarchical relationship between commissioner and provider. By and large, this way of working was at odds with WCMs—referred to as 'the old and new ways of working' (participant 5, case 3, National Health Service (NHS) commissioning). This picture is nuanced though. In both the Falls Prevention Initiative (case 1) and WCM (case 3) interviews, several interviewees stated that while their own preferred way of working aligned with WCM, they found themselves working in organisations that were predominantly, and traditionally, transactional. This created significant challenges for these individuals—who were often the conduits and brokers between WCM and their organisation—as they held a mediating role, trying to reiterate the purpose of WCM to their organisational leaders while simultaneously trying to demonstrate the value of WCM and reassure leaders that their investment in WCM was useful.

**Misaligned understandings and expectations** surrounding WCM were another common barrier mentioned in cases 1 and 3. Several interviewees, especially those who were more peripheral to WCM, stated that they did not understand what WCM was and how it differed from Active Gloucestershire. WCM was seen as intangible by many of these stakeholders, and in part, this was due to what they perceived as complicated messaging. The premise of WCM was not consistently communicated, or understood, by those involved and this contributed to further misunderstanding and confusion. Similarly, if interviewees were predominantly involved in a project (eg, Falls Prevention Initiative or Community-based Initiative) rather than WCM more broadly, they often struggled to see how these particular projects linked with the wider work of WCM. And so, when asked what their role was within WCM, these interviewees either (1) did not see themselves having a role, or (2) did not know what was expected of them. While one of the strengths of Active Gloucestershire, noted by interviewees, was their communication with wider stakeholders, this was not consistent.

The last barrier was that of **COVID-19**. It is important to note that most of the data collection for the second case study, the Community-based Initiative, was undertaken before the pandemic began, and so there was very little data on this in the data set. In the other two cases, COVID-19 meant that much of the planned implementation work was paused or cancelled, and WCM and the Falls Prevention Initiative were required to adapt in response to the imposed restrictions (see strategies section). The primary reasons for this stagnation were that wider stakeholders reverted back to their silos because their organisational priorities changes (ie, the need to 'survive'—participant 7, case 3, local third sector organisation) and many organisations had to furlough staff. This prevented progress being made between stakeholders who were involved in WCM before the pandemic began.

> It's just been such a rubbish year and I think that that has overshadowed and become even more of an obstacle to getting stuff going because you've got the people that are shielding for themselves or others [from COVID-19], the low mood increase, lack of motivation, the disinterest in doing anything because things almost seem pointless… You've got to understand it and it's been impossible, I would say, to get things as good as they could be. – Participant 6, Case 3, Local third sector organisation

### Contextual factors

Contextual factors are those which are not part of the intervention, or were already established preimplementation, but affect its subsequent implementation and impact.[35][36] Interviewees often described the **organisational strengths of Active Gloucestershire**; the organisation was said to be well-connected, respected and a reliable source of knowledge within the county. In short, Active Gloucestershire were already well placed to guide the WCM raft, to be the backbone organisation. That is, the organisation that facilitate, and are accountable for, the implementation (and adaptation) of WCM.

> The result of one of those conversations was the head of commissioning said, 'We need your people off furlough,' and that led to us working over the summer holidays with children that were in social care, have got PTSD, all these things… We would have found our way to that table, but it would have taken a lot of leg work, whereas Active Gloucestershire have got these inroads to these various different partners and have been great at keeping that door open for charities like us. - Participant 8, Case 3, Local third sector organisation

Stakeholders also suggested that WCM was **pushing on a slightly open door**. The council, and several national and local funders, were becoming more receptive to systems approaches. There was already a history of falls prevention work in the county (case 1), a growing interest and application of place-based and asset-based community development (case 2), and a rich and cohesive voluntary sector within the Community-based Initiative area (case 2). Systems approaches have also been adopted in Gloucestershire to other issues such as obesity and ACEs. That said, some organisations involved in case 1 were said to work in a transactional manner whereby Active Gloucestershire are accountable to a commissioner. Organisational engagement in WCM was influenced in part by their institutionalised ways of working. Thus,

linking back to the analogy, Active Gloucestershire had prior knowledge of some of the river conditions before commencing WCM. They also had several crew members, or at least their respective organisations, warmed up and ready to go.

> It's almost like you want to put a heat map over the county, or a magnet, and there are some particular individuals that are really, really interested in systems change. They can see the strategic benefit of it. However, many of those individuals are stuck in traditional institutions and therefore really struggle to innovate, so they will try and be supportive but there'll be certain hoops that they have to jump through or political issues that they're encountering – Participant 3, Case 3, Local third sector organisation

### Impacts and outcomes

We derived four main types of impact from the analysis: (1) physical activity-related impacts, (2) relational impacts, (3) sectoral impacts and (4) null or negative impacts. It is important to note here that the **physical activity-related impacts** were not objectively measured, but rather they were perceived changes in physical activity noted by the interviewees or REM participants. Interviewees—especially in relation to case 2—suggested that the opportunities for physical activity among black, Asian and Minority Ethnic (BAME) women had improved. This was also reported in relation to the older adults (case 1) who received the educational materials from the Falls Prevention Initiative. More broadly though, in case 3 and through the REM analysis, we identified many instances where physical activity, for individuals, communities and populations, may have been influenced.

> They had over 100 women get involved cross all the different activities, and mostly BAME women who wouldn't engage, or who aren't very well connected. So, when the different women come in, they have carried on as well. Over 100 women. That's not bad when you started off with just 12 women who wouldn't engage with other communities. – Participant 1, Case 2, City council

The main impacts, however, as per this analysis, were **relational** and **sectoral**. There was said to be improved cohesion and collaboration between stakeholders involved in, or on the periphery of, WCM, and many examples were given. As previously noted, we found that at least 82 organisations were involved in WCM and that 23 networks were attended or organised. Being part of WCM enabled stakeholders to feel like they were part of something bigger, an initiative or way of working that went beyond their own organisation. This was supported by the generation and dissemination of new funding via WCM—enabling organisations to come together. It was through these new or improved relationships that we saw cross-sectoral impacts; for example, the work of WCM aligned with and supported the work being undertaken

on mental health, air quality, tackling inequality and town planning (among others). And many commented that their own mindset (ie, a way of thinking) and that of their senior leaders had started to change due, in part, to their involvement in WCM. Lastly, through the REM, we found that WCM and physical activity had been integrated into a minimum of six policies across the county, ranging from the Health and Wellbeing Strategy (county-wide) to City Planning policies (city-wide) to Transport Planning policies (city-wide).

> Like I say, there's other people that you see and other ladies and it's finding out what they're doing as well. It's doing things together as a group. They're not only like—it's not to be said, we're going to [steering group member A] and nobody knew about [steering group member B]. It's given [steering group member C] the opportunity to get to know the people in the community in this area. – Participant 9, Case 2, Community member

There were some **negative or null impacts** reported too. Some interviewees stated that there had not been any beneficial impacts on them or their organisation, and that this was due to a lack of engagement or an inability to engage due to COVID-19. This meant that some of the organisations which were involved in WCM were still working, largely, in silos. Other null impacts were more specific to individual case studies, for example, some interviewees said that the Falls Prevention Initiative had not improved the number of older adults being referred to strength and balance classes.

> I thought after the [systems] mapping exercise it was so nice to bring all those organisations together, really beneficial. We had some really positive conversations and I was really hoping there would be some follow-up to that so that I can then start to work with more organisations and look at how we fit as part of this wider approach and not just in silo. I think, for me, that's what We Can Move is supposed to be. It's to bring everyone together and, yeah, create a system change, whereas we're still just cracking on as we're doing. – Participant 7, Case 3, Local third sector organisation

### DISCUSSION

There is currently limited evidence available regarding the implementation of systems approaches in the field of physical activity and across public health more broadly.[27] This paper highlights the key strategies and contextual factors that were associated with the implementation of WCM in Gloucestershire. It also illuminates the types of impact experienced within the first three years of implementation, but notes that there were several barriers which influenced the implementation, and thus, the impact of WCM. Given that there is a paucity of research, this paper provides an important, novel and timely contribution to

the evidence base, with several implications for future practice and evaluation.

## What's needed to start and maintain a systems approach?

Our study identified several important factors to consider when initiating and maintain a systems approach. Understanding, and talking to, the motivations of the wider stakeholders across the system was imperative to kickstarting this work. Active Gloucestershire already had strong relationships across the county, but nevertheless, they had to draw people into WCM. This meant understanding the strategic priorities of other stakeholders and organisations, and in doing so, talking to the shared benefits that could be realised through their involvement in WCM. For example, many organisations were looking to do more partnership work across the county. Active Gloucestershire were able to highlight that a core benefit from engagement in WCM would be that organisations are encouraged to work more closely together, to achieve greater impact and improve the use of scarce resources.

The initial and successful engagement of stakeholders helped WCM to develop the 'right' crew (linked to the rafting analogy). Understanding the initial motivations of the wider stakeholders, and speaking to the shared benefits, brought them aboard. However, once aboard, the role of Active Gloucestershire (as the guide of the raft) was to help stakeholders see the bigger picture—to recognise that their own work and priorities are intertwined with others—and to consider the ways in which they could work together. Formal networking events and meetings, as well as informal relationship brokering, were methods used by Active Gloucestershire to encourage collaboration and a shared vision between stakeholders. These are not new phenomena and are deemed by many to be core ingredients of a systems approach and primary outcomes of systems or adaptive leadership.[26 27 34 43–47] In effect, Active Gloucestershire were developing the crew and encouraging them to paddle together towards a common goal.

Adaptation is one of the main characteristics of a complex adaptive system,[48] and therefore has to be accounted for and anticipated when implementing a systems approach. The impact of COVID-19 caused substantial and rapid adaptations in the local, national, and international systems. For WCM, it meant that many preplanned initiatives had to be paused (eg, engagement with schools), but it also brought about new opportunities for collaborative working which may otherwise not have occurred. Given the flexibility in the design of WCM, it was able to adapt accordingly to these turbulences rather than coming to a halt. It did mean though that some stakeholders, such as the local Public Health team, became less involved (as they were integral to the COVID-19 response) while other organisations, such as local faith groups, became more involved (eg, through distribution of food and activity packs to the community during COVID-19 lockdown restrictions). We referred to

this adaptability of WCM as 'reading the river'. Donella Meadows refers to this as 'dancing with systems'.[49]

## What type of impacts may we expect from a systems approach?

Capturing the diversity of impacts arising from a systems approach is difficult.[50] Within this evaluation, we brought together two methods (REM and semi-structured interviews), which we found helpful in trying to better understand the anticipated and unanticipated impacts of WCM across the Gloucestershire systems—that is, the adaptive and emergent nature of the system and of WCM. REM allowed us qualitatively to capture, close to real time, many of the activities and impacts associated with WCM over a two-year period. The interviews were then used to gather further information about these impacts, and to learn more about how these impacts may have occurred. Our analysis showed that there were perceived changes in physical activity among particular communities (eg, Muslim women within an area of Gloucester—case 2), and changes to the physical activity environment (eg, changing policies around access to leisure facilities and changes to physical activity opportunities in school). However, as others have found,[51] system approaches such as WCM foster new relationships—and cohesion between stakeholders—across the system.

Using the REM, we were also able to demonstrate where WCM benefited other strategic priorities and made intangible impacts, tangible. We noted that WCM was embedded into six policies and had aligned with 22 other agendas (including air quality, town planning, inequalities and mental health). In the 2019 Lancet commission,[24] Swinburn *et al* refer to such initiatives as double or triple duty actions—those which contribute towards several outcomes at once. Linking back to the initial role of Active Gloucestershire, and perhaps other backbone organisations, we were able to see how the appeal of shared benefits as a way of initially engaging stakeholders can be realised through a systems approach. We also documented less tangible, or softer, impacts through REM which included the changing mindsets of senior leaders within the system, enhanced and more cohesive relationships between partners, and the WCM Theory of Change being considered in other areas of the UK. While the relational and mindset aspects of these intangible outcomes are aspired when implementing a systems approach,[23 26 43 50] they are seldom reported on, but here, REM enabled us to surface, capture and analyse these as part of the WCM evaluation.

## What might moderate the implementation and impact of a systems approach?

We highlighted several key factors which influenced the implementation and impact of WCM, and these would likely occur in other areas who are aiming to adopt a systems approach. In line with characteristics of a complex adaptive system,[52] it is important for stakeholders to assess the 'initial conditions' in which the systems approach is to

be implemented.[53 54] Gloucestershire—and many of the organisations within it—was a county which was somewhat primed for WCM; some people (eg, council staff, NHS staff, voluntary and community sector organisations) were familiar with systems approaches, the associated jargon and many had already established cross-sectoral relationships. Similarly, Active Gloucestershire were a well-respected and credible organisation. Linking back to the rafting analogy, WCM started out with a sturdy raft, a strong guide and a forming crew. Some wishing to adopt a systems approach may also have strong foundations to build on, while others may not, and so prospective implementers and evaluators need to be mindful of the starting point and anticipate what impact this may have on early efforts.[35 55 56]

Once WCM commenced, other challenges arose which influenced the rate at which it built momentum. A major challenge was a clash of mindsets, between those wishing to implement—or be involved in— a systems approach and those wanting to retain a more traditional service commissioning/delivery hierarchy. This becomes more problematic if these are institutional or sectoral mindsets (eg, across healthcare) rather than that of a particular individual (eg, a manager within healthcare). Historically in the UK, and elsewhere, public and population health (which include physical activity) has been delivered using a traditional medical model.[19 57–59] In turn, this means that many issues—such as low levels of physical activity—are treated and prevented akin to other medical issues. Thus, this often leads to the funding, commissioning and provision of individual focused services and programmes—for example, physical activity sessions, health promotion campaigns, weight management programmes. These medicalised approaches have been deeply engrained within healthcare and public health, and in part, have contributed to siloed working.[60 61] In this evaluation, despite endorsement of WCM, some stakeholders and organisations appeared uncomfortable with the WCM approach; it meant devolved power, shared responsibility and ownership, less tangible outcomes and accepting uncertainty. These are not findings unique to this context. Crane et al[61] highlighted this as one of four major challenges facing the sustainment of population health programmes, and Alderwick et al[47] noted it as a significant hurdle facing the collaboration between healthcare and non-health care organisations. Many see bold, and systems, leadership as a means of working through these challenges.[43 44 46 58 61]

### Strengths and limitations

This study draws on the data from semi-structured interviews and REM. The latter represents a novel participatory approach to capture the wider impacts of a programme over time. We adapted the method, and report on the findings here, for the first time in the field of systems change in public health. Our robust methodological approach meant that we could gain in-depth insight into the complexity of WCM; through the methods used, the lines of inquiry, the analytical approach, and importantly, the use of an embedded researcher. It is due to these strengths that we believe the findings of this work will be highly relevant and useful to others.

This work is not without its limitations. The impact of the COVID-19 pandemic meant that most data collection was completed online. This likely affected the quality of the REM sessions as it reduced participation and interaction between stakeholders. It also meant that we reduced the length of the sessions, and so it is possible that in-person, longer REM sessions would have provided greater depth to the outputs. Similarly, we only included the implementation team in the REM workshops to help bound our data collection. This limits the ripples within the outputs to those known to the implementation team. In retrospect, and if in person sessions were feasible, we would have included wider stakeholders in REM sessions to examine the impacts towards the periphery of WCM. Finally, this paper draws on two of the seven methods used to evaluate WCM. While the semi-structured interviews and REM findings provide unique, and detailed, insights into WCM, the other methods (not reported on here) allowed us to understand stakeholder engagement (eg, via stakeholder survey), the complexity of physical inactivity (eg, via participatory systems mapping) and participation in events (eg, via secondary analysis of routine data). It was not feasible to include data from all of these sources in this paper, and those interested in the wider evaluation are encouraged to read Nobles et al.[37]

### Future considerations

This research has highlighted several important implications and considerations for those implementing or evaluating systems approaches in Public Health. First is the importance of understanding context. Both implementers and evaluators should spend considerable time understanding the initial conditions within the local systems (eg, organisations, relationships, key actors, local action and policy, historic sociopolitical factors). These conditions—as shown in the results—contributed to the initial implementation of WCM. Second is the need for stakeholders to determine what lies within the scope of the evaluation—acknowledging that different people will value different elements of the systems approach. Where possible, stakeholders should seek to establish the evaluation boundaries and foci early and clearly communicate these to all involved stakeholders. Note too that these boundaries and foci may change over time. Building on this, third is that evaluators may wish to examine the different perspectives of wider stakeholders involved in the approach. Aim to understand their motivations for engagement, what they know of the approach, how they anticipate it to work, and what impacts and outcomes would be of value to them. This may help ensure that the evaluation is meaningful to a range of actors, not just those who

are interested in the primary outcome (ie, increasing population physical activity). A fourth consideration is to question how principles of systems science can be further embedded in the implementation and evaluation of these approaches. For example, how can evaluations gather more information on the adaptive nature of the system, and of systems change efforts? How can implementers ensure—or enhance the likelihood of—identifying and changing deep-rooted leverage points within the system? How can both parties ensure that the evaluation contributes to the continued learning and refinement of the systems approach? And fifth, as others point out,[56 62] further methodological innovation and publication of similar evaluations are warranted to advance the knowledge base around public health-orientated systems approaches.

## CONCLUSIONS

This paper is one of few to report on the evaluation of a physical activity-orientated systems approach. We highlighted several important strategies and contextual factors which enabled WCM to function as it did, and moreover, captured a wide range of impacts, both anticipated and unanticipated, that were associated with its implementation. These findings are important for the field. They present an honest and robust account of a wide range of stakeholder perspectives who were engaged in, or affected by, a systems approach. They highlight some of the barriers experienced when implementing the approach, and it is likely that others will come across these too in the future. This paper advances the evidence base and provides the findings from a novel, participatory method of REM which was triangulated with interview findings. Further research is needed to continue building the evidence base in this area.

**Author affiliations**
¹Population Health Sciences, University of Bristol, Bristol, UK
²National Institute for Health and Care Research Applied Research Collaboration West (NIHR ARC West), University Hospitals Bristol and Weston National Health Service Foundation Trust, Bristol, UK
³Royal Borough of Windsor and Maidenhead Council, Maidenhead, UK
⁴Active Gloucestershire, Gloucester, UK
⁵Centre for Exercise, Nutrition and Health Sciences, School for Policy Studies, University of Bristol, Bristol, UK

**Contributors** JN, SR, RJ and CFos conceptualised the paper, and JN wrote the first draft of the manuscript. All authors (JN, CFox, AI-W, TB, SR, RJ and CFos) provided substantial input into later drafts of the manuscript. All authors refined and discussed subsequent versions of the manuscript before approving the final version. JN is responsible for the overall content as the guarantor.

**Funding** The evaluation of We Can Move was commissioned by Active Gloucestershire. Some of James Nobles', Russell Jago's and Sabi Redwood's time was supported by the National Institute for Health Research (NIHR) Applied Research Collaboration West (NIHR ARC West). The Article Processing Charge was funded through NIHR ARC West.

**Disclaimer** The views expressed in this article are those of the author(s) and not necessarily those of the NIHR or the Department of Health and Social Care.

**Competing interests** None declared.

**Patient and public involvement** Patients and/or the public were not involved in the design, or conduct, or reporting, or dissemination plans of this research.

**Patient consent for publication** Not applicable.

**Ethics approval** Ethics approval was granted by the Faculty of Health Sciences, University of Bristol (Ref. 91145) and all participants provided written informed consent. Participants gave informed consent to participate in the study before taking part.

**Provenance and peer review** Not commissioned; externally peer reviewed.

**Data availability statement** Data are available on reasonable request. Anonymised data are available to bonafide researchers on request.

**ORCID iDs**
James Nobles http://orcid.org/0000-0001-8574-4153
Sabi Redwood http://orcid.org/0000-0002-2159-1482

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
