## [Reviewer comments · BMJ Open]

ARTICLE DETAILS

TITLE (PROVISIONAL)	Navigating the river(s) of systems change: a multi-methods, qualitative evaluation exploring the implementation of a systems approach to physical activity in Gloucestershire, England
AUTHORS	Nobles, James; Fox, Charlotte; Inman-Ward, Alan; Beasley, Tom; Redwood, Sabi; Jago, Russ; Foster, Charlie

VERSION 1 – REVIEW

REVIEWER	Koorts, Harriet Deakin University
REVIEW RETURNED	18-May-2022

GENERAL COMMENTS	Comments Thank you for the opportunity to review this paper, which evaluated implementation and perceived impact of a systems approach to PA in Gloucestershire, using three different case studies of WCM. I congratulate the authors on writing in depth about a complex topic, using language that is accessible and transparent for the reader. The study tackles an interesting and understudied area, which is greatly needed given the increasing swell in systems approaches to health improvement. The paper was enjoyable to read, and provides interesting and important reflections for the field. Recommendations are to improve the reporting of methods and results as this would strengthen the manuscript. Whilst I recognise that the authors have published the methods in detail elsewhere, this is a novel and innovative approach with many steps. I would expect more detail in the current paper to enable a clearer picture of what went on. Firstly, for interpretability of the results, but also so the reader can ascertain if these methods are something they too want to adopt, and could then follow up for specifics in the methods paper. In addition, the study appears to be capturing implementation strategies (ways to promote implementation of an initiative), as opposed to implementation mechanisms (which are causal pathways explaining how strategies work to achieve desired outcomes), so clarity on this is needed. My specific comments are as follows: Abstract 1. There is no mention what the systems approach is targeting (to improve PA). This is needed in the abstract.2. Participants states 31 interviews, yet sample size in the COREQ checklist domain 2 states 32 interviews. Which is correct?3. COVID is reported as a barrier only in the Abstract, but the results and discussion in the main body of the paper describe COVID as an opportunity too. The reporting of this aspect needs to be balanced and transparent in the Abstract.
---

4. Consider including 'implementation' as a key word
5. There is no mention of a framework analysis despite being the analysis approach taken.
6. Realist principles are reported as a strength of the study, but there is little information how a realist approach informed the data collection or analysis, nor is it mentioned in the abstract. Consider revising the strengths listed, or making the role of realist evaluation in this study clearer. I.e. did it inform any data collection instruments etc?

Introduction

7. Line 26 is informative: "Through a systems approach, it aimed to influence the physical activity levels across the Gloucestershire population, with a specific focus on populations with very low levels of activity (less than 30 minutes per week)." This type of additional detail would be beneficial for the abstract as it is currently unclear what the systems approach in Gloucestershire was trying to do.

8. Line 49: The aim in the abstract is 'mechanisms of implementation', yet here it is 'mechanisms of actions when implementing'. This could imply mechanisms of actions that occur in parallel to implementation (but are unrelated). Implementation also includes inaction, and the phrase 'action' implies you are studying the mechanisms of only those events, which 'occurred' as an action. Suggest replacing to 'mechanisms of implementation' for consistency.

9. Recommend including a clear definition of what you mean by a mechanism of implementation (See Lewis 2020 A systematic review of empirical studies examining mechanisms of implementation in health). What you appear to be capturing are implementation strategies (methods used to promote the implementation of an evidence-based practice or program) as opposed to implementation mechanisms (a process or event through which an implementation strategy operates to affect desired implementation outcomes). This needs definition and clarity throughout.

Methods

10. It is unclear how realist principles have informed the study. Do they inform interview/ workshop questions etc? Can you expand how you have applied realist principles in this study?

11. REM is introduced on line 34, but is described in detail much later in the paper. What you have on Line 7 "Ripple Effects Mapping is a participatory, qualitative method for understanding and visualising the wider intended and unintended impacts of a project or programme..." is good but needs to come earlier in the paper.

12. Check reporting of 31 participants with COREQ stating 32

13. Box 1 and Table 1 refer to a theory of change – there is no further detail of what this is? I think this is important so readers wishing to adopt a similar approach can see how a theory of change relates to the systems approach. It is also critical as Table 2 Case 3 states the theory provided a framework for the stakeholders to integrate into, yet is unclear how this would be the case.

Data collection

14. Line 48 and Line 59 refers to REM outputs, but these are abstract as REM has not been described. Suggested describing REM earlier for clarity

	15. Line 50: More information is needed on this annual stakeholder survey – is this something you implemented as part of the project or its's existing in the org? What data did you use from it and why? 16. A time line for data collection or diagram showing the stages of the methodology would be extremely helpful. Or even subheadings to make the stages clearer. For example, you state the interview questions were guided by REM outputs, but REM are described after the interview process. This is confusing for the reader how each step informed the next. 17. Line 14 introduces 'implementation team', this is the first mention (asides from in the COREQ checklist). This needs to be described that there were different types/levels of stakeholder. Also – are the implementation team just invited to workshops and not interviews? Are members of the implementation team for WCM in general, or each case study? Are they different to those you refer to as 'stakeholders' more broadly? This needs to be described more clearly, who the stakeholders were and how they represented the case studies. 18. Line 21 – clarity over the workshops – were they for each case study or everyone combined "The first REM workshops (December 2019) were conducted face-to-face with the whole WCM implementation and programme team, lasting 120-150 minutes" – Are follow up workshops with the same people? Were the participants representing each case studies mixed? Are three the same people representing more than one case study? This is unclear. 19. Who are programme team? How do they differ from implementation team? Are they all going to have equal experience given the different level of their role? 20. Table 2 Reflecting on the impacts, you state how potential interview participants were identified – this is not mentioned in the methods and needs more elaboration – or signposting that it is described elsewhere if you cannot go into depth. Data analysis 21. Line 37 Mentions qualitative data from workshops, but how was this recorded? Were the workshops audio recorded or is this via note taking? 22. As interviews, workshops and analyses were undertaken by the same researcher, this should be acknowledged in the paper (beyond the COREQ checklist). Whilst you report this in the Relationship with participants section, it is not clear in the manuscript what steps were taken to minimise bias. 23. Unclear how the interview and workshop data was synthesised/triangulated to produce 'themes' which led to the final results. Given that these data were collected at different time points, involved different types/roles of participants (implementation team, project team etc) and case studies, a description of this would be helpful. Results 24. Table 3 contains mechanisms of implementation and impact combined. These have been treated separately in the remainder of the manuscript and it is unclear in the data presented, which factors related to implementation and/or impact. I think this needs to be separated or clearly indicated in the table how they relate. Something can relate to implementation effectiveness and be unrelated to program impact, so this distinction is important. 25. It is unclear how you have defined 'mechanism' and why the factors you list in Table 3 are in fact mechanisms, and not just
--	--

	implementation strategies. For example Line 18, 'dissemination via influential stakeholders and residents' is a strategy for dissemination and implementation (see Knoepke 2019 Dissemination and stakeholder engagement practices among dissemination & implementation scientists: Results from an online survey) not a mechanism (see Lewis 2020 A systematic review of empirical studies examining mechanisms of implementation in health). 26. Table 3 – the contextual factors listed give no indication of their direction of influence. I.e., "Active Gloucestershire going through transition period" – is this pos or a neg contextual influence on implementation? Given that each factor cannot be expanded on in the narrative results (for pragmatic reasons), indicating the direction (or bi-directionality) of the contextual influence in this table is needed. 27. Page 15 Line 7 is a clear interpretation of how a mechanism of implementation led to a specific outcome – up until this point the factors described as mechanisms have seemed mostly as implementation strategies 28. Page 19 3.4 Contextual factors – This definition at the start of this section is good, but needs to come much earlier in the manuscript in the methods. Up until this point, it's been unclear how you are defining these aspects. 29. Page 20 3.5 Impacts and Outcomes – as above, this definition on Line 13 is great, but it's unclear what constituted an 'impact' up until this point, yet they are referred to. These all need to be defined earlier in the paper for improved clarity Future considerations 30. The questions for future exploration are really interesting. Can you make any recommendations or suggestions for ways of tackling these questions? I think this would be extremely valuable for readers who may also agree with your suggestions and would benefit from insight as to how to tackle these questions. 31. Whilst the methods you employed combined data across the case studies, are there any reflections you can offer that are distinct to the case studies? Whilst I appreciate the aim was to explore implementation of WCM holistically, these case studies are very different and any unique reflections you have for each one could be very insightful. Limitations 32. Line 31 mentions use of online platform for follow up workshops and Setting of data collection page 35 also refers to this. It would be good to acknowledge and reflect on implications of in person vs online REM using Vensim, and how that might have impacted participant contributions and thus interpretation of your results.
--	---

REVIEWER	Dodd-Reynolds, Caroline Durham University, Department of Sport and Exercise Sciences
REVIEW RETURNED	20-May-2022

GENERAL COMMENTS	I enjoyed reading this manuscript which presents an evaluation of a systems-informed approach to physical activity in Gloucestershire, in England. I think the paper will be of interest to academics working in physical activity, but also to other stakeholders e.g. public health professionals, physical activity commissioners and delivery teams, and other allied health professions whose role might include physical activity prescription within the community.
---

	Minor phrasing/typos: Box 1, final sentence – ‘doing’ should this be ‘do’? General comment on readability: I did have to read the paper several times, as there is a lot contained within, given the three case studies, mixed methods approach and links to other parts of the project published elsewhere. I have made some notes about these points later in the review. Minor point to aid readability – there are a lot of parentheses – could some of these be reduced or incorporated into the main argument? Box 1 (or elsewhere) I think it would be helpful to provide a little more detail on the ‘Theory of Change’ that is referred to throughout the paper – given that this underpins much of WCM’s work. Can you elaborate a little on how this came about and how it informs the wider evaluation? You note that it comprised systems science, bhvr change principles and social movement building – can you be more specific? Methods – Study design: It would be helpful to clarify the distinction between the whole evaluation (cited ref 35 – line 32) versus the data drawn on for this paper – I think ref 35 is to the final evaluation report and ref 36 to the REM methods paper? For example, if this paper includes areas of key importance/interest – why were these selected here? How does this differ to the main evaluation? Methods I think it would be useful to make clear, at the outset of the methods, that the full REM methodological procedures are published elsewhere, rather than later in the section. The paper might explain further how the work was underpinned by realist evaluation principles, if this was indeed the case. Perhaps it was guided rather than underpinned? If the latter, it would be useful to see greater consideration of the theoretical underpinning. How did this align with the adaptive evaluation framework (page 7 line 5)? I think there needs to be clarity around the mixed methods – in particular the distinction between and role of the REM and the interviews – and how these complemented each other. There is some useful explanation in the discussion (section 4.2) which might be highlighted earlier in the paper. What type of mixed methods design was this? E.g. concurrent, sequential, dominant, equal weighted? I don’t feel this is particularly clear in the paper and in places it is quite confusing as to which method the results reflect. If it is truly mixed methods (resulting in mixing of results), then this needs to be made clear. I am struggling somewhat with the many abbreviations – and the different case studies – have to keep returning to table 1 (which provides useful information) to remind myself. Might the case studies be referred (in text) to as Case Study 1,2,3 etc. as per the quotes IDs? Supplementary file 1 – CCOREQ Checklist. A general comment: I think the checklist contains a lot of info that belongs in the methods section of the paper. The checklist would usually acknowledge many of these points and highlight where in the paper these were explained/represented. However perhaps word count is problematic for the main paper? Suggest re-visiting
--	--

	this, however. For example, some of the key info about the lead researcher's relationship with participants – I don't view this as a problem at all, indeed the evaluation would likely not have been able to take place had such a relationship not been established already – so I feel it's important contextual info. Might you include type of qualitative analysis (framework?) under point 9? Point 20 – did field notes aid your analysis of data in any way? Results Were any end-users involved in the workshops – i.e. those people who are the target of the case studies? Discussion I like the discussion although it is long and does repeat some information from the results. If the paper needed to be reduced, this might be an obvious way to do it, leaving space to focus on the theory and interpretation of findings in context of extant literature. Conclusions Novelty of the paper is noted here as involving the REM method – again – was this the primary method – or was this due to the success of mixed methods? Note – figure 1 is not visible in the PDF.
--	--

VERSION 1 – AUTHOR RESPONSE

Reviewer #1:	
Whilst I recognise that the authors have published the methods in detail elsewhere, this is a novel and innovative approach with many steps. I would expect more detail in the current paper to enable a clearer picture of what went on. Firstly, for interpretability of the results, but also so the reader can ascertain if these methods are something they too want to adopt, and could then follow up for specifics in the methods paper.	Many thanks for this. We have now added further detail to these sections of the methods, and have done so in line with the specific comments made by the reviewer (see below). We have been mindful not to add too much given the length of the manuscript at present, but we do hope that what we have added, improves the transparency and interpretability of the method (taken in conjunction with Table 2).
The study appears to be capturing implementation strategies (ways to promote implementation of an initiative), as opposed to implementation mechanisms (which are causal pathways explaining how strategies work to achieve desired outcomes), so clarity on this is needed.	Thank you for raising this. Your comments, here and further on, have really helped to hone our thinking on this matter. We have decided to slightly adjust the language we use in the paper around “mechanisms of implementation” and now refer to these as “implementation strategies” because we believe that this better reflects what it is that we are reporting, in line with your definition here and those used in the papers cited later on. We conflated the two within our manuscript and appreciate the reviewer picking up on this.
Abstract: 1. There is no mention what the systems approach is targeting (to improve PA). This is needed in the abstract. 2. Participants states 31 interviews, yet sample size in the COREQ checklist domain 2 states 32 interviews. Which is correct? 3. COVID is reported as a barrier only in the Abstract, but the results and discussion in the main body of the paper describe COVID as an	1. This has now been added. 2. Thank you for spotting this typo. It is 31 interviewees and we have amended the COREQ checklist to reflect this, and re-checked all references to participant numbers in the paper. 3. This has now been added. 4. Very good recommendation, thank you. 5. Due to the formatting requirements for the abstract, we had to put information about the analytical approach in the design section. We wanted to ensure that this was present, and the “design” section felt most appropriate for this.

opportunity too. The reporting of this aspect needs to be balanced and transparent in the Abstract. 4. Consider including 'implementation' as a key word 5. There is no mention of a framework analysis despite being the analysis approach taken. 6. Realist principles are reported as a strength of the study, but there is little information how a realist approach informed the data collection or analysis, nor is it mentioned in the abstract. Consider revising the strengths listed, or making the role of realist evaluation in this study clearer. I.e. did it inform any data collection instruments etc?	6. Thank you for highlighting this. We have decided to remove reference to the realist elements within this paper. The reason for this is that our application of the principles was very light touch; we wanted to ensure that our evaluation captured information on important contextual factors that may influence the programme, and also to better understand how the programme was seen to function or be implemented – which we now see as “implementation strategies”. We did not try to generate a programme theory, nor did we focus on CMO configurations. The paper itself doesn't lose meaning, transparency or the ability to repeat the study, based upon what is now included and the changes made. We feel that the paper is now clearer due to these amendments and is less likely to confuse readers who are more experienced in realist evaluation and methods.
Introduction 7. Line 26 is informative: “Through a systems approach, it aimed to influence the physical activity levels across the Gloucestershire population, with a specific focus on populations with very low levels of activity (less than 30 minutes per week).” This type of additional detail would be beneficial for the abstract as it is currently unclear what the systems approach in Gloucestershire was trying to do. 8. Line 49: The aim in the abstract is ‘mechanisms of implementation’, yet here it is ‘mechanisms of actions when implementing’. This could imply mechanisms of actions that occur in parallel to implementation (but are unrelated). Implementation also includes inaction, and the phrase ‘action’ implies you are studying the mechanisms of only those events, which ‘occurred’ as an action. Suggest replacing to ‘mechanisms of implementation’ for consistency. 9. Recommend including a clear definition of what you mean by a mechanism of implementation (See Lewis 2020 systematic review of empirical studies examining mechanisms of implementation in health). What you appear to be capturing are implementation strategies (methods used to promote the implementation of an evidence-based practice or program) as opposed to implementation mechanisms (a process or event through which an implementation strategy operates to affect desired implementation outcomes). This needs definition and clarity throughout.	7. Thank you for this recommendation, and we do completely agree with you, however we are really limited by words in the abstract. We have added that it is a PA orientated approach but we feel that is all that we can add here in order to retain the other essential information. 8. Please see our response to point 6. 9. We have defined context on its first reference in the text. As per the response above, we have also changed implementation mechanism to strategy.
Methods 10. It is unclear how realist principles have informed the study. Do they inform interview/ workshop questions etc? Can you expand how you have applied realist principles in this study? 11. REM is introduced on line 34, but is described in detail much later in the paper. What you have on Line 7 “ Ripple Effects Mapping is a participatory, qualitative method for understanding and visualising the wider	10. This has now been removed from the paper – please see our response to point 6 above. 11. We have now edited the sequencing of information around REM so that a brief explanation comes earlier in the methods. 12. Thanks for spotting this inconsistency, and we have ensured that this is changed in the COREQ supplement. 13. Please see our response to reviewer 2 (comment 2).

intended and unintended impacts of a project or programme...” is good but needs to come earlier in the paper. 12. Check reporting of 31 participants with COREQ stating 32 13. Box 1 and Table 1 refer to a theory of change – there is no further detail of what this is? I think this is important so readers wishing to adopt a similar approach can see how a theory of change relates to the systems approach. It is also critical as Table 2 Case 3 states the theory provided a framework for the stakeholders to integrate into, yet is unclear how this would be the case.	
Data collection 14. Line 48 and Line 59 refers to REM outputs, but these are abstract as REM has not been described. Suggested describing REM earlier for clarity 15. Line 50: More information is needed on this annual stakeholder survey – is this something you implemented as part of the project or its existing in the org? What data did you use from it and why? 16. A time line for data collection or diagram showing the stages of the methodology would be extremely helpful. Or even subheadings to make the stages clearer. For example, you state the interview questions were guided by REM outputs, but REM are described after the interview process. This is confusing for the reader how each step informed the next. 17. Line 14 introduces ‘implementation team’, this is the first mention (asides from in the COREQ checklist). This needs to be described that there were different types/levels of stakeholder. Also – are the implementation team just invited to workshops and not interviews? Are members of the implementation team for WCM in general, or each case study? Are they different to those you refer to as ‘stakeholders’ more broadly? This needs to be described more clearly, who the stakeholders were and how they represented the case studies. 18. Line 21 – clarity over the workshops – were they for each case study or everyone combined “The first REM workshops (December 2019) were conducted face-to-face with the whole WCM implementation and programme team, lasting 120-150 minutes” – Are follow up workshops with the same people? Were the participants representing each case studies mixed? Are three the same people representing more than one case study? This is unclear. 19. Who are programme team? How do they differ from implementation team? Are they all going to have equal experience given the different level of their role? 20. Table 2 Reflecting on the impacts, you state how potential interview participants were	14. In line with our response to point 11, this has now been rectified. 15. Thanks for raising this, and we have now added information to this effect. We hope that this is a nice addition to the paper. We realise that we are having to be rather brief with some of the additional information, and this is predominantly to try and not lengthen the paper too much. 16. Thank you for this suggestion. We have now added in a simple paragraph about this at the end of the data collection sub-section. 17. We have now made amendments throughout the paper so that it is clear as to who the implementation team are. We have also further clarified who was invited to the workshops. We also clarify at the beginning of the results which sectors / organisations the interviewees represent. 18. We have now amended this section to add further clarity. 19. This was a typo and thank you for seeing this inconsistency. 20. This point is included, perhaps implicitly, within the first paragraph of the data collection, “b) using the REM outputs”. We have however revised this section of the manuscript so we hope that this is all much clearer now.

identified – this is not mentioned in the methods and needs more elaboration – or signposting that it is described elsewhere if you cannot go into depth.	
Data analysis 21. Line 37 Mentions qualitative data from workshops, but how was this recorded? Were the workshops audio recorded or is this via note taking? 22. As interviews, workshops and analyses were undertaken by the same researcher, this should be acknowledged in the paper (beyond the COREQ checklist). Whilst you report this in the Relationship with participants section, it is not clear in the manuscript what steps were taken to minimise bias. 23. Unclear how the interview and workshop data was synthesised/triangulated to produce ‘themes’ which led to the final results. Given that these data were collected at different time points, involved different types/roles of participants (implementation team, project team etc) and case studies, a description of this would be helpful.	21. We have added to this to state “included data from interviews and the REM outputs”. 22. We have added further information to state that the use of a second researcher in the analysis helped to reduce some of the bias, and the presentation of findings back to the implementation team further aided this (as per the manuscript). We have also added more information, on a related topic, about the embedded researcher role (i.e. relationship between researcher and participants). 23. Thank you for raising this. The findings, rather than the data per se, were triangulated. We analysed the data from the interviews and the REM outputs separately, but we did use the same analytical framework. We have now added further detail to the data analysis section.
Results 24. Table 3 contains mechanisms of implementation and impact combined. These have been treated separately in the remainder of the manuscript and it is unclear in the data presented, which factors related to implementation and/or impact. I think this needs to be separated or clearly indicated in the table how they relate. Something can relate to implementation effectiveness and be unrelated to program impact, so this distinction is important. 25. It is unclear how you have defined ‘mechanism’ and why the factors you list in Table 3 are in fact mechanisms, and not just implementation strategies. For example Line 18, ‘dissemination via influential stakeholders and residents’ is a strategy for dissemination and implementation (see Knoepke 2019 Dissemination and stakeholder engagement practices among dissemination & implementation scientists: Results from an online survey) not a mechanism (see Lewis 2020 A systematic review of empirical studies examining mechanisms of implementation in health). 26. Table 3 – the contextual factors listed give no indication of their direction of influence. I.e., “Active Gloucestershire going through transition period” – is this pos or a neg contextual influence on implementation? Given that each factor cannot be expanded on in the narrative results (for pragmatic reasons), indicating the direction (or bi-directionality) of the contextual influence in this table is needed. 27. Page 15 Line 7 is a clear interpretation of how a mechanism of implementation led to a specific outcome – up until this point the factors	24. In line with the responses above (see our response to the second main, unnumbered comment), we have now applied “implementation strategies” consistently throughout the paper instead of “mechanisms of implementation”. 25. As above. We really do appreciate you drawing this to our attention. 26. Excellent suggestion and please see amendments (not in tracked changes) within this row of the table. 27. Please see our response to point 6 above. 28. Thank you. This has now been defined upon first mention in the methods. 29. As per point 28.

described as mechanisms have seemed mostly as implementation strategies 28. Page 19 3.4 Contextual factors – This definition at the start of this section is good, but needs to come much earlier in the manuscript in the methods. Up until this point, it's been unclear how you are defining these aspects. 29. Page 20 3.5 Impacts and Outcomes – as above, this definition on Line 13 is great, but it's unclear what constituted an 'impact' up until this point, yet they are referred to. These all need to be defined earlier in the paper for improved clarity.	
Future considerations 30. The questions for future exploration are really interesting. Can you make any recommendations or suggestions for ways of tackling these questions? I think this would be extremely valuable for readers who may also agree with your suggestions and would benefit from insight as to how to tackle these questions. 31. Whilst the methods you employed combined data across the case studies, are there any reflections you can offer that are distinct to the case studies? Whilst I appreciate the aim was to explore implementation of WCM holistically, these case studies are very different and any unique reflections you have for each one could be very insightful. Limitations 32. Line 31 mentions use of online platform for follow up workshops and Setting of data collection page 35 also refers to this. It would be good to acknowledge and reflect on implications of in person vs online REM using Vensim, and how that might have impacted participant contributions and thus interpretation of your results.	30. This is a difficult one as we are struggling already for words within this paper, and we are hesitant to add in more where possible. Whilst we do have several thoughts of our own on these considerations, we do not feel as though we have sufficient room to do justice to this. We hope that offering up the considerations as we have enables others to think too about these, without being influenced or constrained by our own thoughts on these matters. 31. This is a very good point and something that we would have liked to do in light of more space. We very much made this our focus within the writing of the results (i.e. pulling out differences between case studies), but we wanted the discussion to focus on the broader issues that come out of our findings, and to place these in the context of the wider literature. 32. Thank you for pointing this out, and we have reflected upon this within the new section on strengths and limitations. Whilst it is not linked directly to this paper per se, we are developing some online training around the REM method, and within that training, we talk to some of the points that you right raise here.
Reviewer #2:	
Minor phrasing/typos: Box 1, final sentence – 'doing' should this be 'do'? General comment on readability: I did have to read the paper several times, as there is a lot contained within, given the three case studies, mixed methods approach and links to other parts of the project published elsewhere. I have made some notes about these points later in the review. Minor point to aid readability – there are a lot of parentheses – could some of these be reduced or incorporated into the main argument?	Many thanks for this point on the readability, and this is something that we were aware of too when writing the paper. We take all of your comments on board and have done our utmost to simplify things where possible, for example by removing parentheses and reducing abbreviations. We are also mindful not to lengthen the paper further, and so some of the amendments made – based on reviewer comments – are concise in nature. As with all papers, there are lots of places where further information or description could be added, and so we have had to make pragmatic decisions throughout. Many of these decisions are also informed by the specific lines of inquiry that we set out in the aims.
Box 1 (or elsewhere) I think it would be helpful to provide a little more detail on the 'Theory of Change' that is referred to throughout the paper – given that this underpins much of WCM's work. Can you elaborate a little on how this came about and how it informs the wider evaluation? You note that it comprised systems science, bhvr change principles and social movement building – can you be more specific?	This is a really helpful point and something that we spent a long time thinking about when drafting the manuscript. We have taken a pragmatic view here to predominantly refer readers to the broader evaluation which discusses the history and detail of the Theory of Change. In the manuscript, and Box 1, we have added a few more sentences which relate to the specific aspects of systems science and behaviour change principles. We hope that these additions, taken in conjunction with the extra information in the evaluation report,

	enable the reader to get a better understanding of this broad theory of change.
Methods – Study design: It would be helpful to clarify the distinction between the whole evaluation (cited ref 35 – line 32) versus the data drawn on for this paper – I think ref 35 is to the final evaluation report and ref 36 to the REM methods paper? For example, if this paper includes areas of key importance/interest – why were these selected here? How does this differ to the main evaluation?	Very good point and we have added further information to help clarify this.
Methods I think it would be useful to make clear, at the outset of the methods, that the full REM methodological procedures are published elsewhere, rather than later in the section. The paper might explain further how the work was underpinned by realist evaluation principles, if this was indeed the case. Perhaps it was guided rather than underpinned? If the latter, it would be useful to see greater consideration of the theoretical underpinning. How did this align with the adaptive evaluation framework (page 7 line 5)? I think there needs to be clarity around the mixed methods – in particular the distinction between and role of the REM and the interviews – and how these complemented each other. There is some useful explanation in the discussion (section 4.2) which might be highlighted earlier in the paper. What type of mixed methods design was this? E.g. concurrent, sequential, dominant, equal weighted? I don't feel this is particularly clear in the paper and in places it is quite confusing as to which method the results reflect. If it is truly mixed methods (resulting in mixing of results), then this needs to be made clear.	Thank you for all of these suggestions regarding the methods section. Several of these have been addressed in our responses to Reviewer 1 (specifically, please see our response to point 6 related to the realist principles). There are several additional things in this that you point towards, specifically the adaptive evaluation framework and the mixed methods. We have provided additional information on the point around the adaptive framework in paragraph 2 of the case study sub-section. It is worth noting here that we chose to present the results from the interviews and REM outputs as these methods were consistently applied across the three case studies. On the second point about the mixed methods, we have referred to this as multi-methods in line with the recommendations from Sparkes (2014) and Anguiera et al. (2018), whereby we have used multiple methods from the qualitative paradigm. That said, we have triangulated the *findings* from these two methods to better answer the research questions. We have added further details throughout the methods, and particularly in the data analysis section, to clarify our approach. We also hope that the additions within the data collection section improve the readers understanding of when REM and interviews were used within the study. It isn't quite as clear cut as sequential, concurrent, equal weighted, but we do hope that our additions now help to address your comment.
I am struggling somewhat with the many abbreviations – and the different case studies – have to keep returning to table 1 (which provides useful information) to remind myself. Might the case studies be referred (in text) to as Case Study 1,2,3 etc. as per the quotes IDs?	Thank you and we appreciate that there are a lot in this. We have tried to remove as many of the abbreviations as we could. We have referred to the case studies by their shortened names (e.g. Falls Prevention Initiative and Community-based Initiative) rather than Case study 1, 2 & 3, as the same issue may persist with readers having to go back to Table 1.
Supplementary file 1 – COREQ Checklist. A general comment: I think the checklist contains a lot of info that belongs in the methods section of the paper. The checklist would usually acknowledge many of these points and highlight where in the paper these were explained/represented. However perhaps word count is problematic for the main paper? Suggest re-visiting this, however. For example, some of the key info about the lead researcher's relationship with participants – I don't view this as a problem at all, indeed the evaluation would likely not have been able to take place had such a relationship not been established already – so I feel it's important contextual info.	Many thanks for this point. In an ideal world, we would like to include more information in the paper related to specifics of the methods. However, as you note, we have tried to keep the word count down in the paper to increase its readability. We hope that the COREQ checklist, and the information contained in it, offers the additional information to readers who are interested in this. Furthermore, including this additional information in the checklist means that we increase the transparency of this study. That said, we take on board your comment about the relationship between the researcher and the WCM implementation team, and have added a point in the paper to that effect. You will also see that we have added more clarity

Might you include type of qualitative analysis (framework?) under point 9? Point 20 – did field notes aid your analysis of data in any way?	to specific aspects of the method in line with the recommendations from reviewer 1. We did not use the field notes to aid our analysis, but we have added more detail about the purpose of the notes in the COREQ checklist.
Results Were any end-users involved in the workshops – i.e. those people who are the target of the case studies?	Only in one of the case studies, the Community Based Initiative, however these participants were involved in co-producing many of the activities that arose within this case study. So they could be seen as both part of the “implementation team” and as an “end user”. We noted this within the introductory paragraph of the results.
Discussion I like the discussion although it is long and does repeat some information from the results. If the paper needed to be reduced, this might be an obvious way to do it, leaving space to focus on the theory and interpretation of findings in context of extant literature.	Thank you for this recommendation. In light of the comments from the editor and the first reviewer, we have made some changes to the discussion (specifically, adding in information on the strengths and limitations of this work)
Conclusions Novelty of the paper is noted here as involving the REM method – again – was this the primary method – or was this due to the success of mixed methods?	Thank you for highlighting this. We hope that our edits to the methods have clarified the relationship between the two methods, REM and interviews.

VERSION 2 – REVIEW

REVIEWER	Dodd-Reynolds, Caroline
	Durham University, Department of Sport and Exercise Sciences
REVIEW RETURNED	30-Jun-2022
GENERAL COMMENTS	I enjoyed reading this revised manuscript. Thanks to the authors for providing such clear responses to reviewers' comments. I think the paper reads very well and the methods, in particular, are much clearer. Minor phrasing issue in final sentence of section 2.4 but likely this will be picked up during copy editing.